# Bone and joint infections due to melioidosis; diagnostic and management strategies to optimise outcomes

**Parvati Dadwal[1,2], Brady Bonner[3], David Fraser[4], Jeremy Loveridge[2,5], Grant Withey[6], Arvind Puri[1], Simon Smith[7], Josh Hanson** [7,8]*

**1** Department of Orthopaedic Surgery, Cairns Hospital, Cairns, Queensland, Australia, **2** James Cook University, Cairns, Queensland, Australia, **3** Department of Orthopaedic Surgery, Logan Hospital, Brisbane, Queensland, Australia, **4** Department of Orthopaedics, Prince Charles Hospital, Brisbane, Queensland, Australia, **5** Far North Queensland Hand Surgery, Cairns, Queensland, Australia, **6** Department of Radiology, Cairns Hospital, Cairns, Queensland, Australia, **7** Department of Medicine, Cairns Hospital, Cairns, Queensland Australia, **8** The Kirby Institute, University of New South Wales, Sydney, New South Wales, Australia

* jhanson@kirby.unsw.edu.au

**Data Availability Statement:** Data cannot be shared publicly because of the Queensland Public Health Act 2005. Data are available from the Far North Queensland Human Research Ethics

## Abstract

### Background

Melioidosis, a life-threatening infection caused by the gram negative bacterium *Burkholderia pseudomallei*, can involve almost any organ. Bone and joint infections (BJI) are a recognised, but incompletely defined, manifestation of melioidosis that are associated with significant morbidity and mortality in resource-limited settings.

### Methodology/principal findings

We identified all individuals with BJI due to *B. pseudomallei* managed at Cairns Hospital in tropical Australia between January 1998 and June 2023. The patients' demographics, their clinical findings and their treatment were correlated with their subsequent course.

Of 477 culture-confirmed cases of melioidosis managed at the hospital during the study period, 39 (8%) had confirmed BJI; predisposing risk factors for melioidosis were present in 37/39 (95%). However, in multivariable analysis only diabetes mellitus was independently associated with the presence of BJI (odds ratio (95% confidence interval): 4.04 (1.81–9.00), p = 0.001). BJI was frequently only one component of multi-organ involvement: 29/39 (74%) had infection involving other organs and bacteraemia was present in 31/39 (79%). Of the 39 individuals with BJI, 14 (36%) had osteomyelitis, 8 (20%) had septic arthritis and 17 (44%) had both osteomyelitis and septic arthritis; in 32/39 (83%) the lower limb was involved. Surgery was performed in 30/39 (77%). Readmission after the initial hospitalisation was necessary in 11/39 (28%), 5/39 (13%) had disease recrudescence and 3/39 (8%) had relapse; 4/39 (10%) developed pathological fractures. ICU admission was necessary in 11/39 (28%) but all 11 of these patients survived. Only 1/39 (3%) died, 138 days after admission, due to his significant underlying comorbidity.

Committee (contact via email FNQ_HREC@health.qld.gov.au) for researchers who meet the criteria for access to confidential data.

**Funding:** The author(s) received no specific funding for this work.

**Competing interests:** The authors declare that no competing interests exist.

## Conclusions

The case-fatality rate from melioidosis BJI in Australia's well-resourced health system is very low. However, recrudescence, relapse and orthopaedic complications are relatively common and emphasise the importance of collaborative multidisciplinary care that includes early surgical review, aggressive source control, prolonged antibiotic therapy, and thorough, extended follow-up.

## Author summary

Bone and joint infections (BJI) are a recognised and life-threatening manifestation of melioidosis but frequently only one component of multi-organ involvement. Almost all patients with BJI due to *Burkholderia pseudomallei* have risk factors that predispose them to developing melioidosis, but diabetes appears to be the most closely associated with BJI. The vasculopathy, neuropathy, structural deformity and decreased immunity that is seen in many individuals with diabetes increases their risk of lower limb infection and these factors–combined with greater exposure to *B. pseudomallei* in the soil and surface water–might explain the lower limb predominance that is seen in many series of melioidosis-related BJI. Patients require early multimodal imaging, appropriate microbiological sampling, early surgical review, prompt source control and an adequate duration of antibiotic therapy–and, in many cases, critical care support–to ensure optimal outcomes. Collaborative, multidisciplinary care reduces the case-fatality rate of melioidosis BJI, but recrudescence, relapse and orthopaedic complications are common in survivors and therefore extended patient follow up is essential.

## Introduction

Melioidosis—the disease caused by the opportunistic, environmental gram negative bacterium *Burkholderia pseudomallei*–is endemic to tropical regions, particularly northern Australia and Southeast Asia. Infection can occur via percutaneous inoculation, inhalation, aspiration, and occasionally ingestion [1]. Melioidosis can affect otherwise healthy individuals, but ~85% have predisposing risk factors for the infection that include diabetes mellitus, hazardous alcohol use, chronic kidney disease, chronic lung disease, malignancy, and immunosuppression [2,3].

Pneumonia is the most common clinical manifestation of melioidosis but almost any organ can be affected, and the disease often involves the skin, the genitourinary tract, the liver or the spleen [2]. Bone and joint infections (BJI) are also a recognised manifestation of melioidosis. In a large, prospective Australian study, 1% and 3% of patients presented with osteomyelitis and septic arthritis, respectively, with an additional 6% subsequently diagnosed with BJI during their illness [2].

Multi-organ involvement is common in melioidosis, and BJI frequently represents just one manifestation of disseminated disease. This is likely to explain why the case-fatality rate of patients with BJI due to *B. pseudomallei* can rise to 37% in low- and middle income countries (LMIC) [4,5]. Even in well-resourced settings, with access to advanced critical care support, a case-fatality rate of 8% has been reported and many survivors experience long-term disability [6,7]. Prompt recognition of the infection is therefore essential and clinicians must have a high level of suspicion for BJI in cases of melioidosis. Meanwhile, patients require early multimodal

imaging, appropriate microbiological sampling, early surgical review, prompt source control and an adequate duration of antibiotics with activity against *B. pseudomallei* to ensure optimal outcomes [8,9].

The incidence of melioidosis in Far North Queensland (FNQ) in tropical Australia has doubled in the last 20 years although the case-fatality rate has fallen significantly over the same period [10,11]. Melioidosis BJI is a common presentation of the disease in FNQ, and outcomes are generally good. We aimed to document the multidisciplinary approach to the management of melioidosis BJI in this well-resourced setting to provide insights for the optimal management of this life-threatening infection.

## Methods

### Ethics statement

The Far North Queensland Human Research Ethics Committee provided ethical approval for the study (HREC/15/QCH/46–977). As the data were retrospective and de-identified, the Committee waived the requirement for informed consent.

Cairns Hospital is a 531-bed, tertiary referral hospital in FNQ and the sole provider for both public orthopaedic and microbiological services in the region. The hospital serves a population of about 290,000 people dispersed across an area of 380,000 km$^2$; approximately 17% of the population identify as Aboriginal or Torres Strait islander Australians (hereafter respectively referred to as First Nations Australians).

We identified all cases of culture-confirmed *B. pseudomallei* infection between January 1$^{st}$, 1998, and June 1$^{st}$, 2023, managed at Cairns Hospital using the state-wide electronic laboratory database AUSLAB. Since October 2016 these data have been collected prospectively. In each case, where possible, the hospital medical record was reviewed and the patients' demographics, comorbidities, their site of infection, surgical management, antibiotic therapy, and clinical course were recorded.

A child was defined as an individual aged ≤16 years. If an individual did—or did not— identify as a First Nations Australian, this was also documented. If individuals lived in the neighbouring Torres and Cape Hospital and Health Service—a region that comprises the Cape York Peninsula and the Torrs Strait Islands—they were said to have a remote residence. Individuals presenting between December 1 and April 30, were said to have a wet season presentation [12]. An individual with less than 2 months of preceding symptoms was said to have an acute presentation; if symptoms had persisted for ≥2 months individuals were said to have a chronic presentation [2]. Pre-existing risk factors for melioidosis—including diabetes mellitus, hazardous alcohol use, chronic kidney disease, chronic lung disease, malignancy, and immunosuppression—were sought and recorded as described previously [13]. If there were insufficient data to determine if a risk factor was not present, it was presumed to be absent. If patients had none of these six predisposing factors, they were said to have no risk factors for melioidosis.

Imaging data including plain radiographs, computed tomography (CT), magnetic resonance imaging (MRI) and positron emission tomography (PET) were collected. Melioidosis-related BJI was defined as septic arthritis or osteomyelitis confirmed by culture of *B. pseudomallei* from joint fluid or bone culture, or intra-operative findings—or a radiology report— consistent with BJI with *B. pseudomallei* isolated concurrently from a microbiological sample from elsewhere in the body. Patients with BJI were then divided into three categories: osteomyelitis alone, septic arthritis alone or both osteomyelitis and septic arthritis. Primary BJI was defined as patients with bone and joint symptoms (such as localised pain and swelling or an inability to weight bear) at presentation; secondary BJI was said to be present if patients lacked

bone or joint symptoms at presentation, complaining instead of symptoms referrable to other systems.

Operative management was defined as an intervention in an operating theatre; details of the procedure were collected from the medical record. Drainage performed by an interventional radiologist was also defined as an interventional procedure, but ward-based, bedside aspirates or debridement were not. The cumulative number of procedures for each case was recorded.

Complications of the BJI, admission to the Intensive Care Unit (ICU) and death attributable to melioidosis were also recorded. Recrudescence was defined as recurrence during the period that the patient was prescribed antibiotic therapy; relapse was defined as recurrence after completion of their prescribed antibiotic therapy. Recrudescence and relapse were defined as either microbiologically confirmed (if there was a positive concomitant culture of *B. pseudomallei*) or as clinically suspected if cultures were negative, but an attending specialist infectious diseases physician and orthopaedic surgeon thought that it was likely and escalated therapy accordingly. Since 2022, whole-genome sequencing has been available to differentiate recurrence and re-infection in patients with melioidosis at our centre.

De-identified data were entered into an electronic database (Microsoft Excel) and analysed using statistical software (Stata version 14.2). Univariable analysis was performed using logistic regression, the $\chi^2$ or the Wilcoxon rank sum test where appropriate. Multivariable analysis was performed using backwards stepwise logistic regression; variables were selected for inclusion in the multivariable model if they had a $p<0.20$ in univariable analysis. Illustrations were created using online illustration templates (BioRender).

## Results

There were 477 patients with culture-confirmed melioidosis during the study period; their median (interquartile range, IQR) age was 54 (42–65) years, 333 (70%) were male, 232 (49%) identified as a First Nations Australian. There were 39/477 (8%) with confirmed BJI, their demographic characteristics and comorbidities are presented in Table 1. Notably, no children were diagnosed with BJI during the study period and no BJI involved prosthetic material. In multivariable analysis of all patients diagnosed with melioidosis, only diabetes mellitus (odds ratio (95% confidence interval): 4.04 (1.81–9.00), p = 0.001) was independently associated with the development of BJI. The median (IQR) glycosylated haemoglobin in the 31 diabetic patients with BJI was 10.3% (9.4–12.8) compared to a figure of 9.8% (7.5–11.9) in the 199 diabetic patients without confirmed BJI (p = 0.053).

Only 10/39 (26%) patients presented with primary BJI and 5 (50%) of these had melioidosis involving other organs concurrently. There were 29/39 (74%) who presented with other manifestations of melioidosis with BJI only diagnosed subsequently (S1 Table). Overall, 32/39 (82%) had a positive culture for *B. pseudomallei* within 48 hours of their admission, most commonly this was a blood culture. In total, 31/39 (79%) were bacteraemic, including 6/10 (60%) with primary BJI.

There were 14/39 (36%) with osteomyelitis, 8/39 (21%) had septic arthritis and 17/39 (44%) with both osteomyelitis and septic arthritis (S2 Table). In the subset of patients with both osteomyelitis and septic arthritis, 13/17 (76%) had adjacent bones and joints involved while 4/17 (24%) had non-adjacent bones involved (S1 Table). Most (32/39, 82%) patients had a lower limb BJI, although there was a concurrent BJI in the upper limb in 4/32 (13%) of these patients. The tibia was the most common site for osteomyelitis while the knee was the most common site of septic arthritis (Fig 1).

**Table 1. Comparison of the demographic characteristics, comorbidities, and clinical course of individuals with melioidosis who did—And did not—Have bone and joint involvement.**

| | All n = 477 [a] | Bone or joint involvement n = 39 | No bone or joint involvement n = 438 | Odds ratio (95% confidence interval) | p [b] |
|---|---|---|---|---|---|
| Age (years) | 54 (42–65) | 52 (42–57) | 55 (42–66) | 0.96 (0.88–1.05) | 0.41 |
| Child ≤16 years | 23 (5%) | 0 | 23 (5%) | - | - |
| Male sex | 332 (70%) | 27 (69%) | 305 (70%) | 0.98 (0.48–2.00) | 0.96 |
| First Nations Australians | 232 (49%) | 27 (69%) | 205 (47%) | 2.56 (1.26–5.18) | 0.009 [c] |
| Remote residence | 171 (36%) | 20 (51%) | 151 (34%) | 2.00 (1.04–3.86) | 0.04 [c] |
| Wet season presentation | 355 (74%) | 25 (64%) | 330 (75%) | 0.58 (0.29–1.16) | 0.13 [c] |
| Acute presentation | 426/471 (90%) | 35 (90%) | 391/432 (91%) | 0.92 (0.31–2.71) | 0.88 |
| Bacteraemic | 332 (70%) | 31 (79%) | 301 (69%) | 1.76 (0.79–3.94) | 0.17 [c] |
| Diabetes mellitus | 237/460 (51%) | 31 (79%) | 206 (49%) | 4.04 (1.82–9.00) | 0.001 [c] |
| Hazardous alcohol use | 171/444 (39%) | 15 (38%) | 156 (39%) | 1.00 (0.51–1.96) | 0.99 |
| Current tobacco smoker | 221/441 (50%) | 21 (54%) | 200 (50%) | 1.18 (0.61–2.28) | 0.63 |
| Chronic lung disease | 95/447 (21%) | 7 (18%) | 88 (22%) | 0.79 (0.34–1.86) | 0.60 |
| Chronic kidney disease | 59/458 (13%) | 3 (8%) | 56 (13%) | 0.54 (0.16–1.81) | 0.32 |
| Immunosuppression | 57/327 (17%) | 6/25 (24%) | 51/302 (17%) | 1.55 (0.59–4.08) | 0.37 |
| Malignancy | 47/446 (11%) | 1 (3%) | 46 (11%) | 0.21 (0.03–1.54) | 0.12 [c] |
| No risk factors for melioidosis | 73 (15%) | 2 (5%) | 17 (16%) | 0.28 (0.07–1.19) | 0.08 [c] |
| Septic shock | 99/443 (22%) | 8/38 (21%) | 91/405 (22%) | 0.92 (0.41–2.08) | 0.84 |
| ICU admission | 118 (25%) | 11 (28%) | 111 (25%) | 1.16 (0.56–2.40) | 0.70 |
| Died | 55 (12%) | 1 (3%) | 54 (12%) | 0.18 (0.03–1.39) | 0.10 |

All numbers are presented as median (interquartile range) or the absolute number (%).

ICU: Intensive care unit.

[a] Retrospective data collection from cases before October 2016 resulted in some missing data prior to this time and, accordingly, a difference in the denominator for some variables.

[b] p value determined using univariable logistic regression

[c] Variables selected for the multivariable analysis

Plain radiographs, CT, MRI, and PET imaging were performed in 36/39 (92%), 11/39 (28%), 28/39 (72%) and 2/39 (5%), respectively. Evidence of a BJI was present on the initial plain radiograph in 2/36 (5%) and on the initial CT in 4/11 (36%). In comparison, initial MRI detected evidence of BJI in all 28 studies: osteomyelitis in 20/28 (71%), septic arthritis in 4/28 (14%) and both osteomyelitis and septic arthritis in 4/28 (14%) (Fig 2). PET-CT imaging was performed in two bacteraemic patients; in one the source was obscure (a CT chest, abdomen and pelvis had been normal) while in the second the patients was persistently bacteraemic despite appropriate therapy with meropenem. In both, the PET scan identified foci consistent with osteomyelitis which was subsequently confirmed with MRI (Fig 3).

Overall, 9/39 (23%) were managed non-operatively with antibiotic therapy alone. Eight of these 9 patients (8/9, 89%) had uncomplicated osteomyelitis. One patient had pelvic osteomyelitis and adjacent sacroiliac joint septic arthritis however surgery was precluded in this case due to difficulty in accessing the sacroiliac joint.

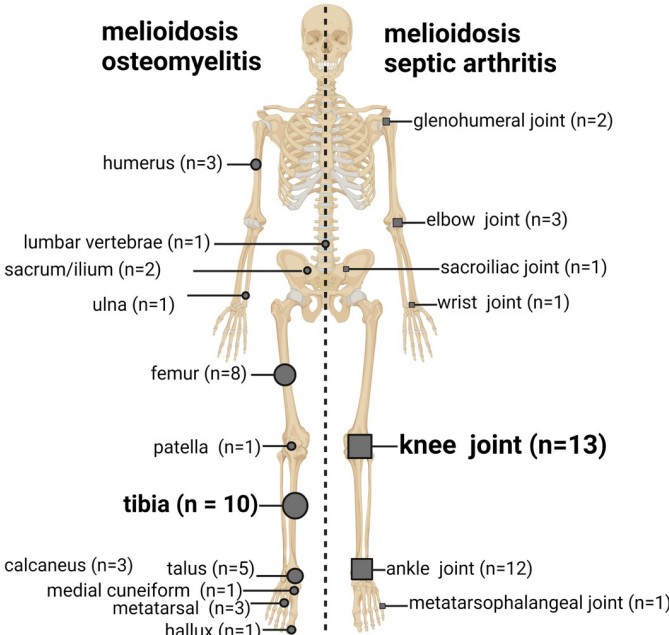

**Fig 1. Anatomical distribution of the osteomyelitis and septic arthritis in the cohort.** n = number of patients. Note: some patients had multiple sites involved. Figure created with BioRender.com.

The remaining 30/39 (77%) had both operative management and antibiotic therapy. The indication for the initial operation was septic arthritis in 23/30 (76%) and osteomyelitis in 7/30 (24%). The initial joint washout for septic arthritis was performed arthroscopically in 14/23 (61%) and as an open procedure in 9/23 (39%). Of the 14 patients who initially had an arthroscopic washout, there were 3 (21%) who subsequently required an open washout. All 7 debridements for osteomyelitis were performed as open procedures.

Patients with osteomyelitis who had surgical intervention had a median (IQR) of 2 (2–3) procedures, patients with septic arthritis who had surgical intervention had a median (IQR) of 2 (1–3) procedures and patients with both osteomyelitis and septic arthritis who had surgical intervention had a median (IQR) of 3 (1–5) procedures. Radiologically guided drainage was performed in 6/39 (15%). This included 4 patients who had radiologically guided drainage in addition to their surgical procedure.

All patients received intensive intravenous antibiotic treatment with ceftazidime or meropenem. The mean (95% CI) duration of total intravenous antibiotic therapy was 6.3 (5.5–7.1) weeks for patients with osteomyelitis, 4.6 (3.3–6.0) weeks for patients with septic arthritis and 6.2 (5.4–7.0) weeks for those with both osteomyelitis and septic arthritis. All patients received trimethoprim-sulfamethoxazole (TMP-SMX) as their initial oral eradication therapy, although 4/39 (10%) developed an adverse drug reaction. This necessitated a change to a second-line agent (amoxycillin-clavulanate or doxycycline) in three patients while one—who was also unable to tolerate either amoxycillin-clavulanate or doxycycline—had extended intravenous meropenem therapy. The mean (95% CI) duration of oral eradication antibiotic therapy was 5.4 (4.5–6.4) months for patients with osteomyelitis, 4.9 (3.6–6.2) months for patients with septic arthritis and 6.2 (4.7–7.8) months for those with both osteomyelitis and septic arthritis.

There were 11/39 (28%) that required admission to ICU; 8/11 (73%) required vasopressors, 6/11 (55%) required intubation and mechanical ventilation and 1/11 (9%) required

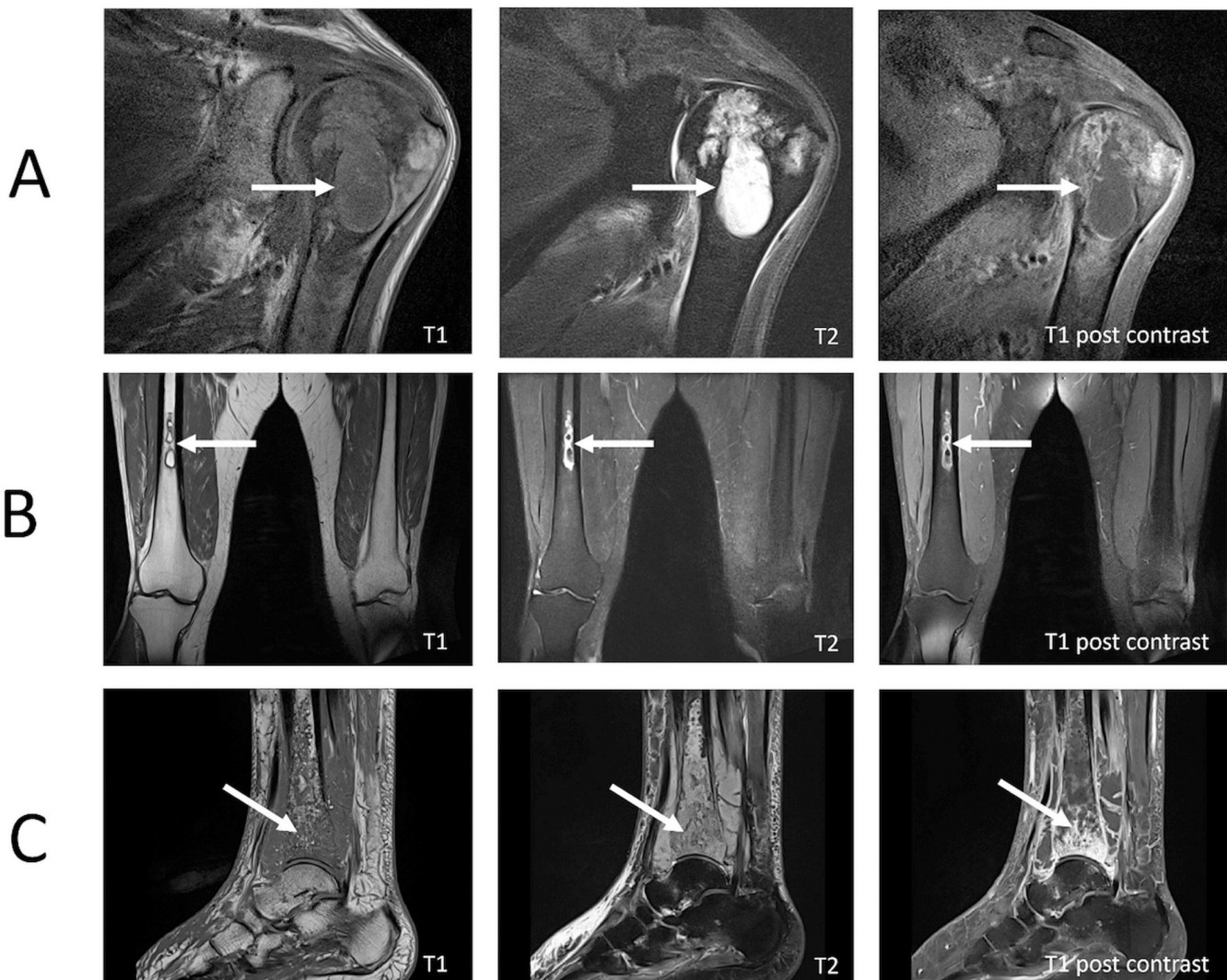

**Fig 2. Magnetic resonance imaging demonstrating osteomyelitis.** (A) Humeral head osteomyelitis and sympathetic joint effusion (B) Mid femoral osteomyelitis with residual abscess (C) Distal tibial osteomyelitis with abscess formation in lateral aspect of distal tibia.

haemodialysis. All 11 patients admitted to ICU survived. Among the 39 patients with BJI, there were 11 (39%) who required readmission; 6/39 (15%) required one readmission and 2/39 (5%), 1/39 (3%) and 2/39 (5%) required two, three and four readmissions respectively. Readmission was usually for a recrudescence of symptoms which required source control, usually repeated surgical drainage of the BJI (S1 Table).

The majority (20/39 (51%) received at least a proportion of their intravenous antibiotic therapy by elastomeric infusion via peripherally inserted central catheter in the hospital's outpatient antibiotic therapy (OPAT) programme; 17/19 (89%) who did not receive intravenous antibiotics in the OPAT programme lived in rural or remote settings where this service was not able to be delivered for much of the study period. The median (IQR) overall length of hospitalisation (which includes the time spent on the OPAT programme) was 39 (21–61) days; there was no difference in length of hospitalisation between the patients who did and did not have operative management (median (IQR): 44 (11–75) versus 39 (23–60) days, p = 0.71).

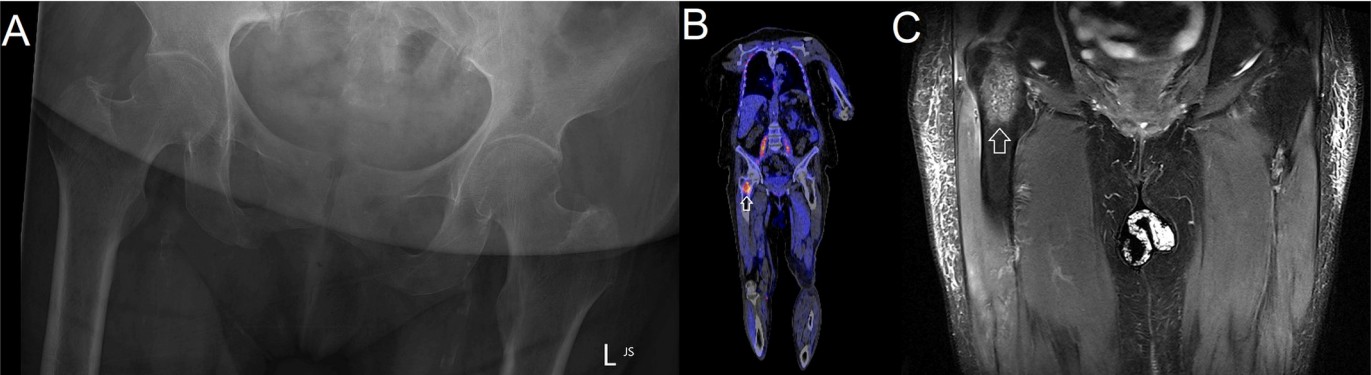

**Fig 3. Imaging findings in patient with osteomyelitis of right femoral neck.** (A) Plain x-ray on day 3 of admission which showing no radiological evidence of osteomyelitis (B) PET-CT imaging on day 9 of admission demonstrating osteomyelitis of right femoral neck (arrowed) (C) MRI imaging on day 13 of admission demonstrating osteomyelitis of right femoral neck (arrowed).

The most common orthopaedic complication of the BJI was pathological fracture (4/39, 10%) (Table 2 and Fig 4). There were 5/39 (13%) who had recrudescence and 3/39 (8%) who had relapse; in all 8 cases recrudescence or relapse occurred at the site of the primary BJI. Microbiological confirmation was possible in 4 (50%), but all 4 of these recurrences occurred before the local availability of whole-genome sequencing, meaning that reinfection could not be completely excluded (although recurrence at the initial site of infection makes this very unlikely). There were 3/5 (60%) episodes of recrudescence and 1/3 (33%) episodes of relapse that were not microbiologically confirmed but were all reviewed by an infectious diseases and orthopaedic specialist who felt that recrudescence or relapse was present (S1 Table). Patients with BJI were more likely to have recrudescence or relapse than patients with melioidosis without BJI during the study period (8/39 (18%) versus 19/438 (4%), p = 0.003).

The sole patient to die in the series was an elderly man with multiple medical comorbidities who presented to hospital after a fall with delirium and community acquired pneumonia. His delirium slowly improved, and he was transferred to the rehabilitation ward to improve his mobility. Eight weeks after his admission he developed fever and ankle pain. His blood cultures grew *B. pseudomallei* and he was diagnosed with osteomyelitis of his talus which was managed non-operatively. His infection initially responded to 6 weeks of intravenous meropenem, but

**Table 2. Complications from melioidosis bone and joint infections.**

|  | Complication | Number (%) |
|---|---|---|
| Orthopaedic. | Pathological fracture | 4 (10%) |
|  | Arthroscopic procedure converted to open | 3 (8%) |
|  | Sinus formation | 2 (5%) |
|  | Bone defect requiring Masquelet procedure | 1 (3%) |
|  | Chronic osteomyelitis | 1 (3%) |
|  | Non-union | 1 (3%) |
|  | Recrudescence | 5 (13%) |
| Infection related. | Adverse reaction to antibiotics | 4 (10%) |
|  | Relapse | 3 (8%) |
|  | Death | 1 (3%) |

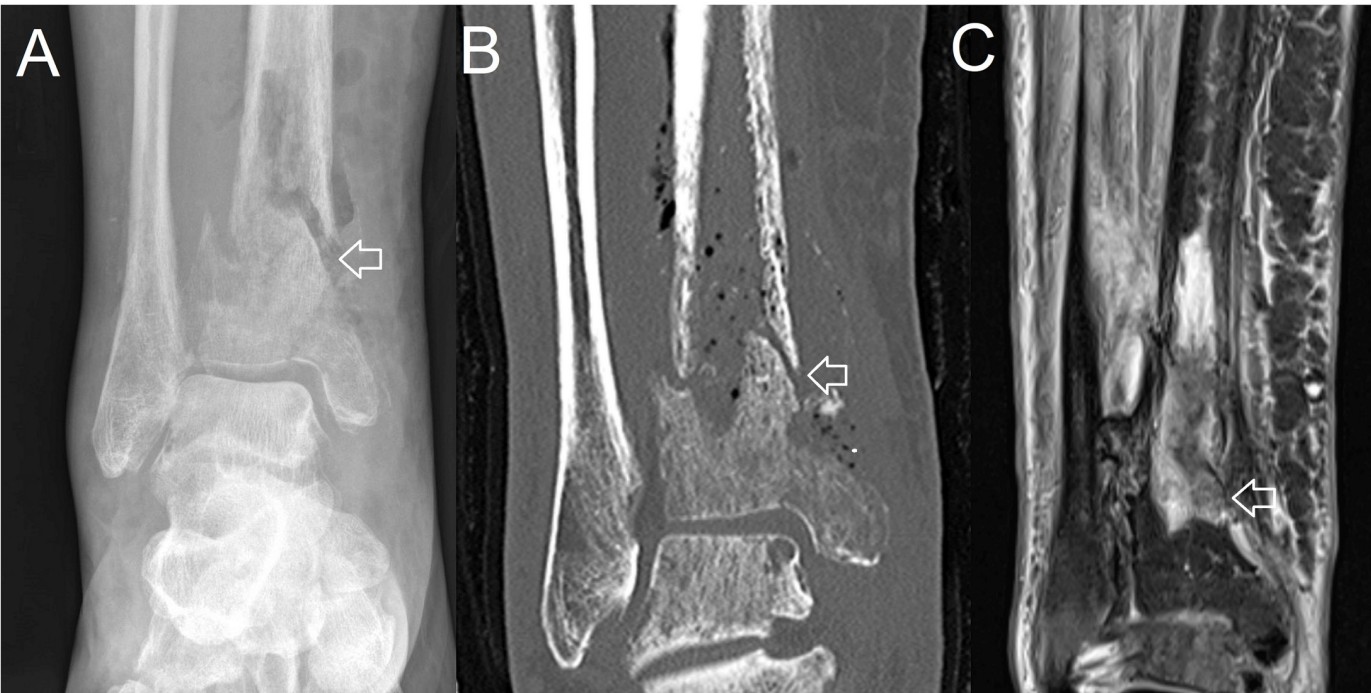

**Fig 4. Pathological fracture of distal tibia.** (A) plain x-ray (B) CT (C) MRI.

he was unable to tolerate either TMP/SMX or amoxicillin/clavulanic acid which caused a rash and vomiting, respectively. The attending infectious diseases team elected to continue his intravenous therapy, however, despite receiving a total of 10 weeks of meropenem, he died 138 days after his admission to hospital from evolving respiratory failure due to his underlying chronic obstructive pulmonary disease, pulmonary fibrosis, and morbid obesity.

## Discussion

BJI is a common manifestation of melioidosis in this region of tropical Australia and is associated with significant morbidity. However, access to advanced imaging and laboratory support and specialist orthopaedic, infectious diseases and critical care services in Australia's well-resourced public health system means that patients with melioidosis and BJI in the region have a very low case-fatality rate. Case-fatality rates of up to 37% have been reported from case series in LMIC settings [4,5] (Table 3). However, although 79% of patients in our series were bacteraemic, and almost 30% required ICU care, there was only a single death, which was explained, predominantly, by the patient's underlying comorbidities. These encouraging results are likely to be explained by local clinicians' early recognition of the disease and prompt empirical antibiotic therapy with activity against *B. pseudomallei.* This, in turn, is enabled by electronically promulgated management guidelines and reliable antibiotic supply chains. Although more than half of the patients resided in remote communities up to 900 kilometres from the hospital, an effective local hub-and-spoke model enabled early transfer to a tertiary hospital where patients were able to access collaborative, multidisciplinary care including sophisticated ICU support, specialist orthopaedic services, aggressive source control and prolonged antibiotic therapy [14,15].

The association between diabetes and BJI in this cohort echoes findings from other series, [4–6,17–19], however, this is the first study to demonstrate that diabetes is independently

**Table 3. Comparison of studies of melioidosis bone and joint infections from the literature.**

| Authors | Year | Study period | Country | n | Died n (%) | SA, OM or both | Lower limb predominance | Proportion with diabetes |
|---|---|---|---|---|---|---|---|---|
| Kosuwon et al. [6] | 2003 | 1997–2000 | Thailand | 25 | 0 | SA | No [a] | 72% |
| Pande et al. [16] | 2011 | 2001–2007 | Brunei | 8 | 0 | Both | Yes | 64% |
| Shetty et al. [7] | 2015 | 1989–2015 | Australia | 50 | 4 (8%) | Both | Yes | 56% |
| Teparrukkul et al. [4] | 2017 | 2012–2014 | Thailand | 74 | 25 (34%) | SA | Yes | 88% |
| Zueter et al. [5] | 2017 | 2008–2014 | Malaysia | 19 [b] | 7 (37%) | Both | Yes | 84% |
| Gupta et al. [17] | 2021 | 2018–2020 | India | 11 | 1 (9%) | Both | No | 82% |
| Wu et al. [18] | 2021 | 2010–2019 | China | 44 | 9 (20%) | Both | Yes | 86% |
| Current study | 2024 | 1998–2022 | Australia | 39 | 1 (3%) | Both | Yes | 78% |

SA: Septic arthritis; OM: osteomyelitis.

[a] In multivariable analysis upper limb involvement was associated with melioidosis BJI

[b] Also included soft tissue infection due to melioidosis; only 9/19 (47%) had bone and joint involvement

associated with BJI in individuals with melioidosis. Diabetes was the most common predisposing factor for melioidosis in the region during the study period, but in multivariable analysis, patients with melioidosis and diabetes were over four times as likely to have BJI than melioidosis patients without diabetes. It is notable that over 80% of the diabetic patients with BJI had poorly controlled diabetes with a glycosylated haemoglobin > 9%. Diabetes results in blunted *B. pseudomallei*-specific cellular responses during acute infection and individuals with diabetes mellitus have a 12-fold higher risk of melioidosis after adjustment for age, sex, and other risk factors [1,20,21]. The vasculopathy, neuropathy, structural deformity and decreased immunity that is seen in many individuals with diabetes increases their risk of lower limb infection and these factors—combined with greater exposure to *B. pseudomallei* in the soil and surface water—might explain the observation that over 80% of BJI in the cohort involved the lower limb, a figure than rose to almost 90% in diabetics [22]. The fact that BJI was seen more commonly in First Nations Australians and individuals living remotely is likely to be explained by the greater burden of diabetes—and higher proportion of First Nations Australians—in the region's remote communities where there is also greater exposure to the pathogen [2,13,23].

It was also notable that no child was diagnosed with melioidosis BJI during the 25-year study period. Children are less likely to develop melioidosis, although life-threatening and even fatal disease can occur [24–26]. The lower incidence of melioidosis in children is explained predominantly by the lower rate of the comorbidities - including diabetes - that predispose individuals to developing the disease. It was also notable that there were no patients who had a BJI involving prosthetic material [19]. A similarly low rate of *B. pseudomallei* BJI involving prosthetic material was noted in a previous Thai study, which was hypothesised to be at least partly due to the lower rate of joint replacement in that country [4]. However, this is assuredly not the case in Australia where the prevalence of joint replacement is 23%, and 13% in those aged >85 years and 65–84 years, respectively [27]. The FNQ region has a population of over 290,000 and joint replacements and orthopaedic implants are performed every day in local hospitals, therefore the absence of a single BJI involving prosthetic material over the >25 years of this study is a striking finding.

There is significant variation in the proportion of patients with melioidosis who develop BJI in the international literature. This is likely to be related to the nature of the study (prospective versus retrospective), the authors of the study (case series reported by orthopaedic surgeons may have a greater focus on BJI) and the definition of BJI employed in the study (was BJI the primary presentation or was it one manifestation of multiorgan involvement). The

landmark Darwin Prospective Melioidosis Series (DPMS) reported that 4% of their patients presented with BJI, with up to 6% developing secondary BJI (3% septic arthritis and 3% osteomyelitis) in the subsequent 3 weeks [2]. This proportion is similar to the proportion of patients with BJI in our cohort and in other high volume Australian centres [28]. Asian series typically report a higher proportion of BJI in their case series [4,6,29], with one Indian study [30] reporting BJI in 48% of their cases. It is unclear if this is the result of differences in the patient cohort (Asian series typically have a far greater proportion of patients with diabetes), differences in exposure history (Asian series often have a greater proportion of agricultural workers) or differences in access to diagnostic support [21]. Geographic variation in the clinical phenotype has been noted with other manifestations of melioidosis, which have been linked to carriage of the $Bim_{Bm}$ virulence gene [31]. While patients infected with *B. pseudomallei* isolates carrying the $Bim_{Bm}$ and *fhaB*3 genes may have a more complicated clinical course [31,32], an association between BJI involvement and individual virulence factors has yet to be established.

Early diagnosis of melioidosis is critical. Most deaths will occur within days of presentation, but even among survivors, delays in diagnosis allow disease evolution, increasing the risk of more complicated disease and long-term sequelae [1,11,33]. In this context it noteworthy that the sole death in the cohort occurred in a patient in whom the diagnosis was delayed until over 6 weeks after his hospitalisation. Clinicians should especially consider the diagnosis in patients presenting with BJI and evidence of infection in other organs, particularly if they have risk factors for melioidosis. In endemic regions this may influence the selection of empirical antibiotic regimens although, of course, other pathogens may present with BJI as a component of disseminated infection [4,34,35]. It should be re-iterated that in most centres, the diagnosis of melioidosis can only be confirmed with culture and this also precludes coinfection [1,36]. Serology is neither sensitive—nor in endemic regions, specific enough—to guide clinical care [1].

Access to sophisticated diagnostic testing facilitated the diagnosis of melioidosis in our cohort, which was also likely expedited by greater local clinical awareness of the infection after a recent doubling of its local incidence [10]. Over 90% of the patients in the cohort with osteomyelitis had an acute presentation and as radiological changes of acute osteomyelitis may not be visible in adults for 10–14 days, it was notable that in many cases, *B. pseudomallei* was isolated in blood cultures before the BJI was diagnosed at a time when plain radiographs and CT imaging were non-diagnostic [37]. Ready access to MRI and PET imaging in the cohort enabled early BJI diagnosis and expedited source control, indeed, 72% of our cohort were able to have MRI imaging and 5% were able to access PET-CT. Clearly, many patients in rural and remote locations—particularly those in LMIC settings—will often not have access to this sophisticated imaging. For clinicians working in these locations repeated symptom review, thorough physical examination, frequent culture and sequential plain imaging may be necessary to establish the diagnosis.

Even if melioidosis is diagnosed, suboptimal treatment increases the risk of early death and of recrudescence and relapse which may be fatal [2,33,38]. In Australia, the rate of relapse is much lower than in other jurisdictions which is explained, at least in part, by the longer courses of intravenous antibiotic therapy prescribed for the infection [39,40]. Australian guidelines recommend that patients with melioidosis and septic arthritis receive 4 weeks of intravenous antibiotics followed by 3 months of oral antibiotics, while it is recommended that patients with melioidosis and osteomyelitis receive 6 weeks of intravenous antibiotics followed by 6 months of oral therapy [41]. However, even though patients in this cohort were prescribed this therapy in a well-resourced universal health system, 13% of patients in the cohort had recrudescence and 8% had disease relapse. This was frequently in the setting of suboptimal adherence to oral eradication therapy which is partly explained by adverse drug reactions to the high dose TMP/SMX prescribed as eradication therapy (S1 Table). Up to 30% of patients prescribed TMP/SMX eradication will require a dose reduction or discontinuation of the

therapy [42,43], and there is therefore interest in high volume centres in reducing the duration of eradication therapy [44]. This might be achieved by extending the duration of intravenous therapy and/or optimising source control [11,39].

Indeed, Australian guidelines for the management of melioidosis emphasise early, aggressive source control, which is facilitated by access to sophisticated imaging and specialist surgical support in the country's well-resourced universal health system [40]. However, despite the significant literature on the appropriate duration of antibiotic therapy in melioidosis, there is very little published data that describes optimal surgical approaches to management of the disease [1,11,45]. The limited number of published series of BJI in patients with melioidosis, the variation in its presentation, the heterogeneity of the affected patients and the diversity of health systems in which they have been managed has precluded the publication of definitive, comprehensive guidelines. However, some principles are clearly important. The fact that BJI was present so commonly in our patients with melioidosis, means that the diagnosis of BJI needs to be considered in all patients with melioidosis and compatible symptoms—especially in those with diabetes mellitus and bacteraemia—as unrecognised osteomyelitis is an important predictor of relapse [39]. Our cohort also shows that when BJI is present it is also often multifocal and that if septic arthritis is present, the adjacent bone is also frequently involved which portends a more complicated course [7].

Well established guidelines for the management of septic arthritis recommend early washout with the aim of reducing bacterial load, removing inflammatory cytokines, and decreasing intraarticular pressure [46]. Meanwhile thorough drainage, extensive debridement of necrotic tissue, appropriate dead-space management, adequate soft-tissue coverage, and the restoration of blood supply are critical to the management of osteomyelitis [7,9]. The optimal surgical approach depends largely on the anatomical location and the patient's suitability for surgery. In the absence of intraosseous collections, sequestrae, involucra, or extraosseous/subperiosteal collections, patients with melioidosis osteomyelitis can be managed with antibiotics alone. However, patients who are not responding to therapy, should be reimaged and, if necrotic bone is identified thorough surgical debridement is usually necessary [7].

The high rate of readmission, recrudescence and relapse seen in our series suggests that thorough aggressive source control and close longitudinal follow up are essential components of care. Almost half of the patients in our cohort had repeated washout or debridement procedures and these sustained efforts to achieve optimal source control are likely to have contributed to the cohort's positive outcomes. In another series of Australian patients with BJI due to *B. pseudomallei*, over 60% of the patients having an operation required multiple procedures and readmission was less common in patients having operative debridement than in those having minor operative procedures [7].

Although our study did not examine BJI due to other pathogens during the study period—precluding the possibility of direct comparison—some characteristics of the clinical course of the patients with BJI due to *B. pseudomallei* bear emphasis. Despite a median of 6 weeks of intravenous antibiotics and 6 months of oral therapy—far longer the antibiotic courses than patients usually receive for BJI [47]–no fewer than 28% of the cohort required readmission and this was frequently for repeated drainages. This is a figure that is similar to the readmission rate of 26% seen in another large Australian cohort of patients with BJI due to *B. pseudomallei* and represents a higher rate than is seen with other pathogens [7]. In one large French study of over 36000 cases of BJI (almost a third of whom had a BJI involving prosthetic material), for instance, *Staphylococcus aureus* was the most commonly identified pathogen and only 19% of cases required readmission [19]. In the OVIVA study, where *S. aureus* was also the most commonly isolated pathogen and which again included patients with infected prosthetic material, patients received a median of 71 days of oral antibiotic therapy or 78 days of

intravenous antibiotic therapy within 7 days of their initial surgery and 17% of patients with complex BJI had treatment failure at 1 year [47]. In contrast, despite a median of 6 weeks of intravenous antibiotics and 6 months of oral therapy and a median of 2 operative procedures, a similar proportion– 18%–of patients with BJI due to *B. pseudomallei* in our cohort developed relapse or recrudescence within 1 year of their presentation.

The study has many limitations. Its retrospective nature precluded comprehensive data collection, especially for some patients managed early in the study period whose medical records were inaccessible. The study may underestimate the prevalence of BJI in our cohort as, particularly earlier in the study period, patients may have died from their infection prior to BJI symptoms developing or before they were able to have sensitive diagnostic imaging. The documentation of operative procedures was not standardised which hindered comparison of different surgical strategies. A study period of over 25 years enabled the identification of more patients with BJI, however it coincided with a significant evolution in the understanding of the optimal management of melioidosis and advancements in supportive care. This inevitably impacted the management of the patients and their clinical course and limits our analysis. Although this is one of the larger published series of BJI due to *B. pseudomallei*, it is a retrospective series of fewer than 40 patients. These patients had a variety of clinical presentations, a range of comorbidities and they were managed by different clinicians, prohibiting the generation of firm conclusions about optimal management strategies. Long-term functional outcomes are also not reported. The patients were managed in Australia's well-resourced universal health system limiting the generalisability of our findings to resource-limited settings. However, a high index of suspicion for melioidosis in the appropriate clinical context, aggressive—and, where necessary, repeated—source control and prolonged antibiotic therapy with extended follow-up are almost certainly likely to be equally relevant in these locations. This extended follow is also critical to ensuring the delivery of comprehensive longitudinal care that also addresses the patients' underlying comorbidities (such as poorly controlled diabetes) and which is also critical to optimal long term outcomes [48].

Given the high reported rates of readmission, recrudescence and relapse, future studies might examine whether there are any differences in the surgical management of BJI due to *B. pseudomallei* compared to BJI due to other pathogens. Ideally these studies would be prospective and would examine BJI due to *B. pseudomallei* and BJI due to other pathogens concurrently. These studies might assess different surgical approaches and then determine the optimum duration of antibiotic therapy. Such studies might be able to determine if adequate surgical source control enables the use of shorter courses of antibiotic therapy for BJI due to *B. pseudomallei*, or, if not, what strategies might improve adherence to the long courses of eradication therapy that are presently recommended [39].

## Conclusions

BJI is a common presentation of melioidosis, especially in individuals with diabetes mellitus. Early recognition, prompt surgical review, thorough debridement, prolonged antibiotic therapy and, frequently, advanced critical care support are necessary for optimal outcomes. However, readmission, recrudescence and relapse appear common—even in high volume centres—and may necessitate frequent, repeated source control and extended follow up.

## Supporting information

**S1 Table. Demographic and clinical characteristics, management, and clinical course of the 39 individuals with bone and joint infections due to *B. pseudomallei*.**
(DOCX)

**S2 Table. Characteristics of the individuals with bone and joint infections due to** *B. pseudomallei*, **stratified by clinical phenotype.**
(DOCX)

## Acknowledgments

The authors would like to acknowledge the many health workers involved in the care of the patients in this cohort.

## Author Contributions

**Conceptualization:** Simon Smith, Josh Hanson.

**Data curation:** Parvati Dadwal, Brady Bonner, David Fraser, Grant Withey, Simon Smith, Josh Hanson.

**Formal analysis:** Parvati Dadwal, Josh Hanson.

**Investigation:** Parvati Dadwal, Simon Smith, Josh Hanson.

**Supervision:** Jeremy Loveridge, Arvind Puri, Simon Smith, Josh Hanson.

**Visualization:** Josh Hanson.

**Writing – original draft:** Parvati Dadwal, Brady Bonner, Josh Hanson.

**Writing – review & editing:** Brady Bonner, David Fraser, Jeremy Loveridge, Grant Withey, Arvind Puri, Simon Smith, Josh Hanson.

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
