## [Decision Letter · Decision Letter 0]

6 May 2024

Dear Dr. Hanson,

Thank you very much for submitting your manuscript "Bone and joint infections due to melioidosis; diagnostic and management strategies to optimise outcomes." for consideration at PLOS Neglected Tropical Diseases. As with all papers reviewed by the journal, your manuscript was reviewed by members of the editorial board and by several independent reviewers. In light of the reviews (below this email), we would like to invite the resubmission of a significantly-revised version that takes into account the reviewers' comments. 

We cannot make any decision about publication until we have seen the revised manuscript and your response to the reviewers' comments. Your revised manuscript is also likely to be sent to reviewers for further evaluation.

Sincerely,

Joseph M. Vinetz

Section Editor

Joseph Vinetz

Section Editor

Reviewer's Responses to Questions

**Key Review Criteria Required for Acceptance?**

**Methods**

-Are the objectives of the study clearly articulated with a clear testable hypothesis stated?

-Is the study design appropriate to address the stated objectives?

-Is the population clearly described and appropriate for the hypothesis being tested?

-Is the sample size sufficient to ensure adequate power to address the hypothesis being tested?

-Were correct statistical analysis used to support conclusions?

-Are there concerns about ethical or regulatory requirements being met?

Reviewer #1: Comments as below

Reviewer #2: (No Response)

Reviewer #3: Study population well defined. Retrospective study, goal to assess outcomes 

Study does not lend itself to a hypothesis

No concern about ethics: Data were retrospective and de-identified, the Committee waived the requirement for informed consent.

Reviewer #4: (No Response)

**Results**

-Does the analysis presented match the analysis plan?

-Are the results clearly and completely presented?

-Are the figures (Tables, Images) of sufficient quality for clarity?

Reviewer #1: Comments as below

Reviewer #2: (No Response)

Reviewer #3: Images and figures well done

Reviewer #4: (No Response)

**Conclusions**

-Are the conclusions supported by the data presented?

-Are the limitations of analysis clearly described?

-Do the authors discuss how these data can be helpful to advance our understanding of the topic under study?

-Is public health relevance addressed?

Reviewer #1: Comments as below

Reviewer #2: (No Response)

Reviewer #3: Limitations clearly outline - retrospective study, small number of patients affected and no standardized documentation of operative strategy

Reviewer #4: (No Response)

**Editorial and Data Presentation Modifications?**

Reviewer #1: Comments as below

Reviewer #2: (No Response)

Reviewer #3: Accept

Reviewer #4: (No Response)

**Summary and General Comments**

Reviewer #1: This is a well written retrospective review of Bone and joint infections (BJI) caused by Burkholderia pseudomallei , in Far North Queensland. It covers about 25 years and a comparative analysis is provided against those cases of infection with this organism where there was no evidence of a BJI.

The main concern that I have with this study is that it compares a very uncommon cause of a BJI (39 cases over 25 years – 1-2 cases a year) with other patients with melioidosis. It really should have compared this cause of a BJI, with the far commoner cause of a BJI – namely Staph aureus. This is a significant limitation of the study. As it is, it offers little that is new. What is needed is how infection of bone or joint with Burkholderia pseudomallei, differs both epidemiologically and clinically from that caused by Staphylococcus aureus (at the very least).

Other points are

Line 156: This is too vague, either the organism was re-isolated and a decision made to restart treatment (relapse) or, there was failure to clinically improve during therapy which was then prolonged. This should not be called recrudescence unless the organism was reisolated during treatment.

Line 168 The results section could be reduced significantly as data is repeated and could be referred to in the tables.

Line 263 Once again this is of little value if it depends on clinical opinion. If there is something unusual about either relapse/recrudescence of a BJI with melioidosis as opposed to say Staph aureus, then it should be mentioned and referenced. If not then this statement should be restricted to cases where the organism was reisolated and proven to be identical.

Line 323 How many of all Melioidosis cases had a prosthetic joint that was NOT infected? If unknown then this statement cannot be made

Line 386 and 396 Is this unique to B.pseudomallei?

 -

Reviewer #2: This is an extremely well written case series involving 39 patients with melioidosis and bone and/or joint involvement. Excellent outcomes were achieved, which would not necessarily have been the case in many less well-developed settings, and I think that both this and the vital importance of early surgical referral could have been emphasised more strongly.

Otherwise, I have only a couple of suggestions.

1. Was genotyping done in the 3 patients with culture-positive recurrence, to confirm labelling the 2 post-treatment recurrences as relapse as opposed to reinfection (although the latter is extremely unlikely). This should be made clear.

2. Figure 1 should be labelled ‘ulna’ rather than ‘ulnar’.

Reviewer #3: This is a very well written manuscript. The authors link risk for meliodosis to presence of diabetes, but given median hemoglobin a1c of 10.3 I perhaps the comment should be about poorly controlled diabetes. May also comment on high rates of tobacco use and more clearly define "hazardous" alcohol use. Do the authors want to comment on why plain radiographs performed so poorly and why there is a strong male predominance?

Reviewer #4: The manuscript briefs the clinical presentation and outcomes of BJI in Melioidosis infected patients . Although well written the paper does not have any novel findings to report , and the number of cases are too less to conclude anything…

PLOS authors have the option to publish the peer review history of their article (what does this mean?). If published, this will include your full peer review and any attached files.

Reviewer #1: No

Reviewer #2: No

Reviewer #3: No

Reviewer #4: Yes: Tushar Shaw
---

## [Editor Report · Decision Letter 1]

24 Jun 2024

Dear Dr. Hanson,

We are pleased to inform you that your manuscript 'Bone and joint infections due to melioidosis; diagnostic and management strategies to optimise outcomes.' has been provisionally accepted for publication in PLOS Neglected Tropical Diseases.

Best regards,

Joseph M. Vinetz

Section Editor

Joseph Vinetz

Section Editor

---

## [Editor Report · Acceptance letter]

29 Jun 2024

Dear Dr. Hanson,

We are delighted to inform you that your manuscript, "Bone and joint infections due to melioidosis; diagnostic and management strategies to optimise outcomes.," has been formally accepted for publication in PLOS Neglected Tropical Diseases.

Best regards,

Shaden Kamhawi

co-Editor-in-Chief

Paul Brindley

co-Editor-in-Chief
